# Effects of (–)-Loliolide against Fine Dust Preconditioned Keratinocyte Media-Induced Dermal Fibroblast Inflammation

**DOI:** 10.3390/antiox10050675

**Published:** 2021-04-26

**Authors:** Ilekuttige Priyan Shanura Fernando, Mawalle Kankanamge Hasitha Madhawa Dias, Dissanayaka Mudiyanselage Dinesh Madusanka, Hyun-Soo Kim, Eui-Jeong Han, Min-Ju Kim, Min-Jeong Seo, Ginnae Ahn

**Affiliations:** 1Department of Marine Bio-Food Sciences, Chonnam National University, Yeosu 59626, Korea; shanura@chonnam.ac.kr; 2Department of Food Technology and Nutrition, Chonnam National University, Yeosu 59626, Korea; 198807@jnu.ac.kr (M.K.H.M.D.); 198793@jnu.ac.kr (D.M.D.M.); iosu5772@naver.com (E.-J.H.); alswn1281@nate.com (M.-J.K.); 3National Marine Biodiversity Institute of Korea, 75, Jangsan-ro 101-gil, Janghang-eup, Seocheon 33662, Korea; gustn783@mabik.re.kr; 4Freshwater Biosources Utilization Bureau, Bioresources Industrialization Support Division, Nakdong-gang National Institute of Biological Resources (NNIBR), Sangju 37242, Korea

**Keywords:** fine dust, human dermal fibroblast, (-)-loliolide, matrix metalloproteinases, inflammation

## Abstract

At present air pollution in parts of East Asia is at an alarming level due to elevated levels of fine dust (FD). Other than pulmonary complications, FD was found to affect the pathogenesis of ROS-dependent inflammatory responses via penetrating barrier-disrupted skin, leading to degradation of extracellular matrix components through the keratinocyte-fibroblast axis. The present study discloses the evaluation of human dermal fibroblast (HDF) responses to FD preconditioned human keratinocyte media (HPM) primed without and with (-)-loliolide (HTT). HPM-FD treatment increased the ROS level in HDFs and activated mitogen-activated protein kinase-derived nuclear factor (NF)-κB inflammatory signaling pathways with a minor reduction of viability. The above events led to cell differentiation and production of matrix metalloproteinases (MMP), increasing collagenase and elastase activity despite the increase of tissue inhibitors of metalloproteinases (TIMP). Media from HTT primed keratinocytes stimulated with FD indicated ameliorated levels of MMPs, inflammatory cytokines, and chemokines in HDFs with suppressed collagenase and elastase activity. Present observations help to understand the factors that affect HDFs in the microenvironment of FD exposed keratinocytes and the therapeutic role of HTT as a suppressor of skin aging. Further studies using organotypic skin culture models could broaden the understanding of the effects of FD and the therapeutic role of HTT.

## 1. Introduction

Harmful effects of fine dust (FD) on the skin are one of the most debated titles in the recent past. Increased air pollution not only affects the respiratory tract but results in systemic toxicity causing cardiac dysfunctions, aggravated asthma, pregnancy complications, congenital disabilities, skin allergies, and inflammatory reactions [1,2,3]. FD is a heterogenous suspension of solid and liquid particles in the air that varies in size and composition. The chemical composition of FD includes sulfates, nitrates, elemental carbon, organic compounds such as polycyclic aromatic hydrocarbons (PAH), biological entities such as allergens, endotoxins, microbes, cell fragments, and metallic ions/elements (e.g., lead, iron, copper, zinc, nickel, and vanadium) [2,4]. 

Epidemiological data on ambient air pollution describes several major skin diseases, including atopic dermatitis, acne, and psoriasis [3]. A recent mouse in vivo study demonstrates that FD can penetrate the skin through hair follicles and sweat ducts, and more severely, the barrier disrupted skin resulting in dermal inflammation and neutrophil infiltration [5]. FD induces the production of reactive oxygen species (ROS), increasing oxidative stress with subsequent production of inflammatory cytokines, chemokines, and other chemical mediators. Fibroblasts are an abundant cell type in the dermis with their primary function of secreting extracellular matrix components that withhold skin integrity. Dermal fibroblasts in the skin reside intimately connected to the upper layer of epidermal keratinocytes. Other than fibroblasts, the dermis contains cells such as macrophages that belong to the immune system, nerves, blood vessels, secretory glands, and hair follicles [6]. Recent literature has shown that FD exposure can adversely affect skin keratinocytes by increasing intercellular ROS production and inflammatory responses, and its long-term effects can lead to the deterioration of the skin barrier [7,8,9]. Inflamed keratinocytes secrete cytokines that affect cells, such as HDFs, in their microenvironment, increasing oxidative stress, inflammation, and the production of matrix metalloproteinases (MMPs), which are involved in the degradation of connective tissue components in the skin [8].

Early studies describe the use of monolayer cultures of in vitro skin cells such as keratinocytes, melanocytes, and fibroblasts as a model for analyzing skin diseases. Although these studies formed the foundation of current knowledge, they are not suitable to imitate the interactions between different cell types, especially the spatial association between skin cell layers. At present, organotypic epidermal skin models are commercially available for in vitro experiments [6]. These skin tissue equivalent models offer significant advantages over monolayer or co-culture systems. However, considering the high cost and materials required to engineer 3D organotypic models, alternative techniques would come in handy as prior models of in vivo evaluations. The integrated culture model of keratinocytes and fibroblasts previously presented [8] is a convenient way of analyzing the mediatory role of keratinocytes on fibroblasts. This method provides the ability to monitor the transfer of signals from keratinocytes to fibroblasts. However, this integrated co-culture technique is not a putative method as it lacks several important cues of fibroblast and keratinocyte environment, such as extracellular matrix (ECM), stiffness, much lower cell density compared to the use of in vitro and organotypic skin culture models.

Antioxidant and anti-inflammatory drugs have shown the potential for treating FD-induced skin diseases [3]. (6S,7aR)-6-hydroxy-4,4,7a-trimethyl-5,6,7,7a-tetrahydrobenzofuran-2(4H)-one also known as (-)-loliolide (abbreviated as HTT) is a monoterpene lactone renowned for its antioxidant and anti-inflammatory effects [7,10]. Recently HTT was isolated from an ethanolic extract of edible brown alga *Sargassum horneri* as one of its active constituents [11]. As an extension of our previous study [7], the present study evaluates the human dermal fibroblast (HDF) responses to HaCaT cell preconditioned media (HPM) primed with FD (HPM-FD) without and with HTT treatment.

## 2. Materials and Methods

### 2.1. Materials

FD (NIES CRM No. 28) was obtained from the National Institute for Environmental Studies (Ibaraki, Japan). Dulbecco’s Modified Eagle Medium (DMEM), Ham’s F-12 nutrient mix, Fetal bovine serum (FBS), and penicillin-streptomycin mixture were purchased from GIBCO INC., NY, USA. Ethidium bromide, acridine orange, 3-(4,5-dimethylthiazol-2-yl)-2,5-diphenyltetrazolium bromide (MTT), 2′ 7′-dichlorodihydrofluorescein diacetate (DCF-DA), Triton™ X-100, Hoechst 33342, and paraformaldehyde were obtained from Sigma-Aldrich Co (St. Louis, MO, USA). SuperSignal™ West Pico PLUS Chemiluminescent Substrate was purchased from Thermo Fisher Scientific (Rockford, IL, USA). Ace-α-^®^ cDNA synthesis kit was purchased from ReverTra (Toyobo, Osaka, Japan). PCR primers were purchased from Bioneer Inc. (Daejeon, South Korea). Primary and secondary antibodies for western blot analysis, goat serum, Prolong^®^ Gold AntiFade Reagent with DAPI, and DyLight™ 554 Phalloidin were bought from Cell Signaling Technology (Beverly, MA, USA).

### 2.2. Cell Culture and Treatment

HaCaT cells were received as a kind donation from the Surface Science Laboratory of Center for Anti-aging Molecular Science, Korea Advanced Institute of Science and Technology. Culture, maintenance of HaCaT cells, FD stimulation, and sample treatment were carried out following the previously reported method [7]. Briefly, HaCaT cells were seeded for 24 h and treated with HTT concentrations of 50, 100, and 200 µM for 2 h. Next, the cells were stimulated with 150 µg mL^−1^ of FD for 2 h. The wells were then emptied, washed twice with new media, re-filled with new media, and incubated for 24 h. After HaCaT cell preconditioned media (HPM) from each treatment group was collected, syringe filtered, and stored at -80 °C until further experiments. Table 1 indicates the treatment strategy and abbreviation used to designate each preconditioned media group.

HDF fibroblasts (ATCC^®^ PCS20101™) were maintained in DMEM media supplemented with 25% F-12, 10% FBS, and 1% penicillin/streptomycin mixture. Subculturing of cells was carried out once every 5 days, with fresh culture media being replaced once every two days. Cells indicating an exponential growth between passages 3-6 were seeded in 96 well cell culture plates (1 × 10^4^ cells per well) and used for experiments. Seeded cells were maintained under 37 °C for 24 h in a humidified incubator. Wells were emptied, and wells except the control were stimulated with preconditioned media from HaCaT cells, which were either nontreated, stimulated with FD or FD, and HTT. An equal amount of fresh DMEM culture media was treated to the control wells. The cells were then incubated for 2 h for stimulation. After the incubation period, the wells were again emptied, washed once, and filled with new DMEM + F12 culture media to remove HaCaT cell-derived cytokines and other chemical mediators which would otherwise interfere with analyzing HDF-derived chemical mediators. Cells were further incubated for 24 h, and the viability was measured by MTT assay [12]. The absorbance of DMSO dissolved formazan crystals was measured using a Molecular Devices SpectraMax M2 microplate reader (Sunnyvale, CA, USA).

### 2.3. Measuring Intracellular ROS Production

HDFs were stimulated in the same way as mentioned above. However, the stimulated cells, after replacing with new media, were incubated for 2 h duration. The fluorescence probe dye DCF-DA that signifies intracellular ROS was treated to the cells and incubated for 20 min. The fluorescence intensity was measured by fluorometry, fluorescence microscopy (EVOS FL Auto 2 Imaging, Thermo Fisher Scientific, Rockford, IL, USA), and flow cytometry (CytoFLEX, Beckman Coulter, Brea, CA, USA). During flow cytometry, cells were harvested by trypsinization before the analysis. FSC vs. SSC gating was applied for eliminating cell debris.

### 2.4. JC-1 Assay

The mitochondrial membrane potential of HDF cells was assessed by JC-1 assay employing flow cytometry and fluorescence microscopy. Fibroblast stimulation follows the same procedure as described above. Cells were harvested 6 h after the media replacement by trypsinization and during the assay, maintained in DMEM + F12 media at 37 °C to avoid mitochondrial depolarization. The assay procedure is following the manufacturer’s guidelines.

### 2.5. Nuclear Double Staining

Nuclear double staining using ethidium bromide and acridine orange mixed stain was employed for detecting apoptotic cells according to our previous method [13]. In brief, 24 h after the media replacement, fibroblasts were treated with the mixed stain (final concentration 2 µg mL^−1^ each) and incubated for 10 min. The wells were emptied, washed with PBS, and re-filled with PBS. Images were taken by an EVOS FL Auto 2 Imaging microscope.

### 2.6. Western Blot Analysis

HDFs were stimulated in the same way as mentioned above. However, the stimulated cells, after replacing with new media, were incubated for 30 min duration for the western blot analysis of upstream mitogen-activated protein kinase (MAPK) and nuclear factor kappa-light-chain-enhancer of activated B cells (NF-κB) molecular mediators. Nuclear and cytoplasmic extracts of cells were obtained by NE-PER^®^ kit (Rockford, IL, USA). Lysates were standardized for protein (40 μg) and loaded to 10% sodium dodecyl sulfate-polyacrylamide gels. After electrophoresis, resolved protein bands were blotted onto nitrocellulose membranes. Membrane strips based on the molecular weight of antigens were separated into chambers and blocked with 5% skim milk in TBST. After, the membrane strips were incubated with primary (1:1000) and HRP-conjugated secondary antibodies (1:3000), respectively. Protein bands were visualized after ECL staining by a western blot imaging system (Davinch-ChemiTM, Core Bio, Seoul, Korea) [14]. Relative intensities of western blot bands were quantified using ImageJ 1.52a software with relevant normalization markers.

### 2.7. Immune Fluorescence Analysis

The method of analysis is described in our previous publication [12]. Briefly, HDF cells were seeded in chamber slides for 24 h. HDFs except the control group were stimulated with preconditioned media from HaCaT cells, which were either nontreated, stimulated with FD or FD, and HTT for 30 min. Next, the wells were PBS washed and treated with 4% paraformaldehyde for a 15 min fixation. Cells were dually PBS washed and incubated for 1 h in normal goat serum (5%) and Triton X-100 (0.3%) in PBS blocking buffer. Cells were incubated with anti-NF-κB p65 antibody, PBS washed and again incubated with Anti-Mouse IgG conjugated with Alexa Fluor^®^ 488. After PBS rinse, slides were mounted with Prolong^®^ Gold antifade reagent containing DAPI and coverslipped. The slides were analyzed by an EVOS FL Auto 2 Imaging microscope.

### 2.8. RNA Isolation and PCR Analysis

TRIzol was used for the isolation of total RNA. cDNA was synthesized by normalized total RNA using a ReverTra Ace-α-^®^ cDNA synthesis kit according to the manufacturer’s instructions. The cDNA prepared after the initial denaturation at 94 °C for 3 min was amplified by 30 PCR cycles with 1 min, annealing at 72 °C, 1 min extension. The list of primers and their sequences is per Table 2. The RT-PCR product was electrophoresed on 1% agarose gels with 0.5 µg mL^−1^ ethidium bromide and visualized under a UV transilluminator (Park et al., 2019a). Relative intensities of expression levels were quantified using ImageJ 1.52a software with GAPDH for normalization.

### 2.9. Intracellular Collagenase and Elastase Activity Assay

Cells were seeded in culture dishes. After stimulation and 24 h incubation, wells were PBS washed, trypsinated, and harvested. Cells were PBS washed and lysed with 0.1 M Tris–HCl buffer (pH 7.6) containing 1 mM PMSF and 0.1% Triton-X 100 by sonication in ice (<4 °C). The protein content in the lysate was quantified using a Pierce BCA protein assay kit. Cellular collagenase and elastase activities were determined according to the Suganuma et al. (2010) method [15].

### 2.10. Statistical Analysis

Experiments were carried out in triplicate (n=3), and numerical values are presented as mean ± standard deviation. One-way analysis of variance was carried out by the statistical software using PASW Statistics 19.0 (Chicago, IL, USA) followed by Duncan’s multiple range test to compare mean values with significant differences of *p* < 0.05 “*”and <0.01 “**”.

## 3. Results

### 3.1. Preconditioned Media from FD-Stimulated Keratinocytes Primed with HTT Attenuated Cell Viability and ROS Production in HDFs

The FD sample used during this study (NIES CRM No. 28) mainly consists of particles having a diameter of less than 15 µm (Figure 1A) with the majority being less than 2.5 µm as reported by Mori, et al. [16]. The EDX analysis indicated that FD contains high amounts of elements such as C, O, Na, Mg, Al, Si, S, Cl, K, Ca, Ti, and Fe (a high amount of carbon is due to the carbon tape). A complete specification of its composition including weight percentages of elements and amounts of polycyclic aromatic hydrocarbons (PAHs) is provided in the product sheet as well as by Mori, Ikuko, et al. (2008) [16]. Previous studies have revealed that FD-stimulation of HaCaT keratinocytes increases intercellular ROS production and, thus, cytotoxicity [7]. There the protective response of HTT was dose-dependent, which reduced cellular oxidative stress while restoring viability. In the present analysis, further studies were conducted to assess the effect of stimulated keratinocytes on fibroblasts in their microenvironment. Furthermore, studies were conducted to assess how HTT primed FD stimulated keratinocytes would roleplay in ameliorating detrimental cellular responses in HDFs. HPM-FD treatment promptly increased intracellular ROS levels compared to the control and HPM treated groups (Figure 1B), with a reduction of cell viability. Effects were recovered when the preconditioned media was obtained from HTT treated HaCaT cells. The increase of HDF viability seen in HPM treated cells could be due to the presence of fibroblast growth factors in the media released by keratinocytes [8]. The fluorescence microscopic (Appendix A) and flowcytometric (Figure 1D) analysis of DCF-DA stained cells indicated an increase of intracellular ROS level for the HPM-FD treatment compared to the control. A minor increase of green fluorescence was observed in HPM treated HDFs compared to the non-treated control group. ROS levels in HDFs were reduced when preconditioned media was obtained from HTT primed FD-stimulated HaCaT cells, anti-parallel to HTT dose. Collectively above evidence confers HTT’s ability to reduce the keratinocyte stimulation plays a role in suppressing intracellular ROS levels in HDFs. Caspases are among the crucial mediators of apoptosis [17]. Caspase-3 is among the most frequently activated protease that catalyzes the cleavage of numerous cellular proteins critical in maintaining cellular homeostasis. Hence analyzing its levels provide vital evidence regarding the loss of cell viability. According to Figure 1C, both control and HPM groups indicated the same level of cleaved caspase-3. A minor increase in cleaved caspase-3 level was observed upon the treatment of HPM-FD. Treatment of preconditioned media from HTT primed HaCaT cells indicated a dose-dependent reduction of cleaved caspase 3.

### 3.2. Preconditioned Media (HPM-HTT-FD) Treatment Attenuated Mitochondria Depolarization and Fibroblast Differentiation

The disruption of active mitochondria is a distinctive feature of the early stages of apoptosis. This includes changes in the mitochondrial membrane potential. JC-1 is a probe dye that undergoes potential-dependent accumulation in mitochondria emitting green (~529 nm) and red (~590 nm) fluorescence consecutively for its monomeric and aggregated forms. The decrease in red/green fluorescence intensity ratio indicates depolarized mitochondria. Herein, JC-1 assay was conducted both by flow cytometry (Figure 2A) and fluorescence microscopy (Appendix A). The majority of control cells showed a higher red fluorescence, while the majority of HDFs treated with CCCP, a commercial mitochondrial membrane disruptor, showed a higher green fluorescence. HPM-FD treatment of the HDFs showed minor mitochondrial damage, which was suppressed when treated with preconditioned media obtained from HTT primed, FD-stimulated HaCaT cells (HPM-HTT-FD). Nuclear staining with ethidium bromide and acridine orange was conducted to assess apoptosis (Figure 2B). None of the HDFs indicated apoptosis, but an altered cellular morphology based on the treatment type was observed. Treatment of HPM increased the HDF density compared to the control, whereas HPM-FD caused a prominent alteration of cell morphology. Altered morphology was recovered when treated with HPM-HTT-FD with correspondent to increasing HTT doses.

### 3.3. Activation of MAPK Derived NF-kB Inflammatory Signaling Was Suppressed by Preconditioned Media (HPM-HTT-FD)

NF-κB and MAPK are two of the critical upstream pathways responsible for the onset of inflammatory responses [8]. Based on western blot analysis, HPM-FD treatment compared to the control increased the phosphorylation of MAPK and NF-κB mediators, while increasing nuclear translocation of NF-κB p65 (Figure 3A,B). HPM-HTT-FD treatment with correspondent to increasing HTT doses suppressed phosphorylation of MAPK and NF-κB mediators as well as the nuclear translocation of NF-κB p65. Nuclear translocation of NF-κB p65 was evaluated by immunofluorescence analysis (Figure 3C). Being consistent with the western blot results, HPM-FD treatment increased the nuclear translocation of NF-κB p65 compared to the control. Treatment of HPM-HTT-FD suppressed the NF-κB p65 nuclear translocation.

### 3.4. Preconditioned Media (HPM-HTT-FD) Treatment Downregulated Inflammatory Cytokines and Chemokines

The expression levels of key inflammatory cytokines were assessed by PCR. According to Figure 4, HPM treatment resulted in a slight upregulation in cytokine (IL-1β, -6, -8, -33, and TNF-α) expression compared to the control. HPM-FD treatment upregulated cytokine expression compared to the control. A subsequent downregulation of cytokines was seen for the HPM-HTT-FD treatment with correspondence to the HTT dose.

### 3.5. Connective Tissue Degradation Was Suppressed by Preconditioned Media (HPM-HTT-FD)

MMPs are zinc-dependent endopeptidases involved in regenerating normal tissues. However, their aberrant increase of expression in dermal fibroblasts is associated with ECM degeneration. Generally, the two major structural proteins of the ECM (type I and type III collagen) are degraded by MMPs-1, 8, and 13 [18]. In addition, the coordinating action of MMP-2 and MMP-9 have been identified as contributing factors to ECM degeneration, photoaging, and tumor progression. As denoted in Figure 5A, HPM treatment slightly upregulated the expression of MMPs (-1, -2, -3, -8, -9, and -13) compared to the control. The expression was further increased upon HPM-FD treatment compared to the control. MMP expression indicated a reduced level in HPM-HTT-FD treated HDFs with correspondence to the content of HTT. MMP activities in fibroblasts are regulated by numerous proteins. Among them are the tissue inhibitors of metalloproteinases (TIMPs) [19]. The balance between TIMPs and MMPs is crucial for maintaining the physiological functionalities of ECM. Per transcriptional analysis, a minor increase was seen for TIMP1 and TIMP2 in HPM-FD treated group compared to the control. HDFs treated with HPM-HTT-FD indicated a reduction of TIMP expression along with the HTT dose. Intracellular collagenase and elastase activity indicated a slight upregulation with HPM treatment (Figure 5B,C). HPM-FD treatment indicated a prompt increase of collagenase and elastase activity, whereas the activity gradually suppressed with HPM-HTT-FD treatment in accordance with HTT dose.

## 4. Discussion

The development of multidisciplinary research, cell culture methods, clinical requirements for the replacement of lost skin tissue, and the regulatory requirements to replace animal models with substitute testing methods have led to the development of integrated co-culture and organotypic models of human skin. In general, these skin in vitro forms consists of dermal matrixes of keratinocytes cultured over fibroblasts. The functionality of these cells depends on their microenvironment [6]. The major role of the skin is to act as a physical barrier of the body, protecting internal organs from external aggressors. Special attention has recently been paid off on research in cutaneous biology and cosmetics science [20]. FD has recently been identified as a risk factor for many serious health conditions due to its effects on various organ systems, including the respiratory, cardiovascular system, and the skin [1,2,3].

FD used in the present study (NIES CRM No. 28) is sourced from a ventilating system of a building located in Beijing, China. The collection method, certified and reference values of elemental composition, size distribution, the mass fraction of polycyclic aromatic hydrocarbons (PAHs), and information of its stability and homogeneity are specified by Mori, Ikuko, et al. (2008) [16]. CRM No. 28 is widely used to study oxidative stress and inflammatory responses in various cells and animal models [12,21,22].

According to our previous studies, FD exposure promptly augmented the ROS level, apoptotic body formation, DNA damage, and pro-inflammatory cytokine and chemokine production in HaCaT keratinocytes altering the production of skin hydration factors, which are essential to maintain skin barrier integrity [7,12]. The present integrated culture technique was employed, hypothesizing that these dysregulated cytokines may trigger inflammatory responses in cells in their microenvironment, such as HDFs. According to the present outcomes, the said FD-stimulated keratinocyte preconditioned media treatment augmented the intracellular ROS level. The results of each analysis method involving fluorometry, flow cytometry, and fluorescence microscopy were comparable with each other. In particular, flow cytometry analysis is considered a reliable method as it measures the fluorescence intensity of individual cells. The employed forward vs. side scatter (FSC vs. SSC) gating excludes the fluorescence occurring from cell debris [12]. The intracellular ROS levels were recovered when the preconditioned media were obtained from HTT treated FD-stimulated HaCaT cells. These results imply that HTT was effective in suppressing the stimulation of keratinocytes and thereby the HDFs resulting in a lowering of intracellular ROS levels.

Accumulated evidence suggests that uncontrolled production of ROS may lead to mitochondrial-mediated apoptosis [12]. Under physiological conditions, the potential of the mitochondrial membranes remains at a slightly higher level. The decreased mitochondrial membrane potential is a key indicator for the detection of apoptosis. Per the present results, the mitochondrial membrane potential had a moderately lower impact in HPM-FD treated HDFs. In comparison, the decrease in mitochondrial membrane potential for the same treatment was moderately lower than the increase in intercellular ROS levels. The loss of mitochondria membrane potential recovered back to normal when the preconditioned media were obtained from HTT treated, FD-stimulated HaCaT cells. Nuclear double staining based on ethidium bromide and acridine orange was used to identify HDFs susceptible to apoptosis. Interestingly, there was no indication of apoptosis for HPM-FD treated HDFs, despite their significantly higher ROS level. Although it was not the expected outcome of the nuclear double staining method, HDFs indicated a noteworthy change in cell morphology on HPM-FD treatment. According to previous studies, morphological changes are attributed to cell differentiation. This alteration observed in cellular morphology returned to the normal level when treated with HPM-HTT-FD. Collectively, these evidences imply that FD exposure of keratinocytes has a low probability of inducing apoptosis in underlying fibroblasts despite the production of ROS. Cleaved caspase-3 is a hallmark of apoptosis. Its extensive involvement in mediating chromatin condensation and DNA fragmentation plays a vital function during normal development while unwanted cells are committed to dying [17]. However, dysregulated caspase-3 cleavage is undesirable as it leads to the loss of cell viability. Results indicated a similar level of cleaved caspase-3 in both the control and HPM treated group. The minor increase of cleaved caspase-3 levels in HDFs treated with HPM-FD relates to the reduction of cell viability in the same group.

Cellular inflammatory responses play a vital role in maintaining the cells/tissue homeostasis against harmful stimuli such as toxic compounds, pathogens, damaged cells, or irradiation by initiating the healing process [23]. However, uncontrolled and recurrent inflammatory responses contribute to the pathogenesis of inflammatory diseases. Inflammatory stimuli such as microbial products, cytokines, and chemical irritants initiate intracellular signaling pathways, which activate the production of inflammatory mediators. Some of the renowned receptors that initiate inflammatory responses are toll-like receptors, receptors for IL-1 (IL-1R), and IL-6 (IL-6R), the TNF receptor, and [23,24]. These receptors trigger the activation of key intracellular upstream signaling molecules, such as nuclear factor kappa-B (NF-κB), mitogen-activated protein kinase (MAPK), and Janus kinase-signal transducer and activator of transcription pathways [23]. Previous studies have shown that HaCaT cells produce inflammatory cytokines after exposure to FD [8]. These inflammatory mediators play a crucial role in influencing the crosstalk between keratinocytes and fibroblasts. Therefore, cytokines in preconditioned media may activate MAPK and NF-κB mediators in HDFs. Hence it is clear that the reduction of cytokines in FD-stimulated HaCaT cells on HTT treatment, which was reported in our previous study [7], contributes to the suppression of upstream inflammatory pathways in HDFs. Immunofluorescence analysis provides a clear comparison of NF-κB p65 nuclear translocation when treated with HPM-FD and its suppression after HPM-HTT-FD treatment corresponding to increased HTT dose.

Inflammatory mediators in HDFs play a pivotal role in mediating the structural integrity of the extracellular matrix of the skin by regulating the transcription of MMPs, collagenase, and elastase [25]. Keratinocytes primed with FD enhanced the HDFs transcription of IL-1β, -6, -8, -33, and TNF-α. However, expression levels were suppressed when treated with HPM-HTT-FD. According to Borg et al. (2013), inflammatory cytokine production by skin cells plays a central role during the manifestation of skin aging and inflammatory diseases [26]. High concentrations of TNF-α increase the synthesis of collagenase while inhibiting collagen synthesis. Irreversible damage occurs due to persistent levels of TNF-α by inducing the production of MMP-9, which causes collagen degradation [27]. Another study indicates that IL-6 and TNF-alpha are implicated in the regulation of TIMP-1 and MMP-1 mRNA expression [19]. The above scenario takes place due to the TNF-α-induced binding of NF-κB transcription factors to the MMP-9 gene [27]. According to Oh et al., regulated activator protein-1 (AP-1) activation and NF-κB mediate the expression of MMP-1, -3, and -9 in HDFs [28]. Transcriptional activity of AP-1 is regulated by c-Jun N-terminal kinases (JNK), extracellular signal-regulated kinase (ERK), and p38-mediated phosphorylation of c-Jun and c-Fos, which form the AP-1 complex by heterodimerization [28,29]. Therefore, MAPK proteins play a pivotal role in ECM degradation. Keratinocytes primed with FD-induced the phosphorylation of NF-κB and MAPK mediators. Moreover, it stimulated the nuclear translocation of NF-κB p65. Suppression of nuclear factor of kappa light polypeptide gene enhancer in B-cells inhibitor (IκB)α, p65, JNK, ERK, and p38 phosphorylation in HDFs indicated that media from HTT primed keratinocytes stimulated with FD might downregulate the transcription of MMP-1, -3, and -9. Furthermore, being upstream, signaling molecules, NF-κB, and MAPK pathways regulate the production of pro-inflammatory cytokines [8]. Hence HPM-FD treatment would create a continuous loop of cytokine-mediated activation of NF-κB and MAPK pathways, aggravating inflammation in HDFs.

MMPs are calcium-dependent zinc-endopeptidases, which under physiological conditions can digest extracellular matrix components, such as collagen, elastin, fibronectin, laminin, and basement membrane glycoproteins [29]. MMPs play a critical role not only in the physiological degradation of ECM, mediating morphogenesis, repair, and angiogenesis of tissues, but also in chronic inflammation, wrinkling, arthritis, osteoporosis, periodontal disease, and tumor invasion as characterized by excessive ECM degradation. In recent years, there has been a growing focus on the exploration of compounds that can inhibit MMP activation in extracellular space as an approach to prevent skin aging. Per the present analysis, a notable difference was observed for the expression of MMP-1, -2, -3, -8, -9, and -13 in HDFs treated with preconditioned media HPM-FD. Media from HTT primed keratinocytes stimulated with FD downregulated the transcription of MMPs in HDFs, providing evidence of HTT’s protective effects against the activation of keratinocytes with subsequent suppression of inflammatory and ECM degradation responses in HDFs.

In addition to cytokines, MMP production in fibroblasts is regulated by numerous proteins including tissue inhibitors of metalloproteinases (TIMPs) and growth factors [19]. TIMPs play a pivotal role as inhibitors of MMPs. Regulating the balance between TIMPs and MMPs is crucial for the physiological functionalities of ECM. As explicitly described by Arpino et al. (2015), TIMP1 is a potent inhibitor of many MMPs excluding MMP-14, -15, -16, -19, and -24 [30]. The function of TIMP2 is related to the inhibition of MMP-2. However, controversial findings suggest that MMP-2 could be an activator of pro-MMP-2. A minor increase was seen for TIMP1 and TIMP2 transcription in HPM-FD treated group compared to the control. It is reported that the regulation of TIMP expression is independent of MMPs. Though the function of TIMP1 and TIMP2 are related to the inhibition of MMP, the increased activity of collagenase and elastase suggest that the activity of MMPs has surpassed the inhibitory effects of TIMPs. HDFs treated with preconditioned media obtained from HTT primed, FD-stimulated HaCaT cells (HPM-HTT-FD) indicated a reduction of TIMP expression along with the HTT dose. According to previous studies cytokines such as IL-1β, IL-6, and TNF-α positively regulate the selective expression of certain MMPs and TIMPs [19]. Similar outcomes were observed during the present analysis where the fluctuation of cytokines (IL-1β, -6, -8, -33, and TNF-α), MMPs, and TIMPs expression was parallel to each other. Herein translational analysis of TIMP and MMP levels together with type 1 collagen would provide a sound conclusion of the ECM degradative effects of FD preconditioned keratinocyte media on dermal fibroblasts and the effects of HTT.

HTT used during the present analysis was isolated from an ethanol extract of the edible alga, *S. horneri*, which is renowned for its ethnopharmacological significance and beneficial health properties including antioxidant, anti-inflammatory, anti-allergic, astringent, antipyretic, and vasodilatory activities [7,11,31,32]. Chemically HTT is a norisoprenoid hydroxylactone, initially discovered from *Lolium perenne* in 1964 [33]. Since then, HTT has been discovered from numerous animal and plant species, including the red ant, *Solenopsis invicta*, and the brown algae *Sargassum crassifolium* [31]. According to Grabarczyk et al., HTT has been discovered from many medicinal plants [31]. Per the present analysis, HTT treatment of FD stimulated keratinocytes roleplay in ameliorating intracellular ROS level and inflammatory responses in HDFs. The effects could be attributed to HTT’s ability to suppress oxidative stress and inflammatory responses.

## 5. Conclusions

Based on clinical evidence and recent findings from diseased tissues, FD is a risk factor for negative skin effects such as atopic dermatitis, eczema, and skin aging [34]. Apart from air quality management, understanding symptoms of skin damage for early diagnosis and exploring the therapeutic potential of drugs has therefore become imperative. Investigations need to be widened to identify sources and harmful components in circulating airborne particles in different areas and to better understand their harmful health effects. Present evaluations focus on the transmission of FD-caused skin aging effects in the keratinocyte-HDF axis and studied the therapeutic potential of the naturally occurring compound, (−)-loliolide. In light of our previous studies and present findings, HTT possesses the potential to suppress FD-induced keratinocyte inflammation and thereby abate HDFs inflammation and degradation of ECM components. Further studies are required to assess the efficacy of HTT in FD stimulated in vivo models similar to the one described by Jin Seon-Pil, et al. (2018) [5]. Identifying major biomarkers that are associated with FD-induced inflammatory responses would be helpful to design diagnostic and treatment strategies. With further evaluations in place, HTT can be implemented as a drug candidate for pre-clinical research and the development of functional ingredients in cosmetics.

## Figures and Tables

**Figure 1 antioxidants-10-00675-f001:**
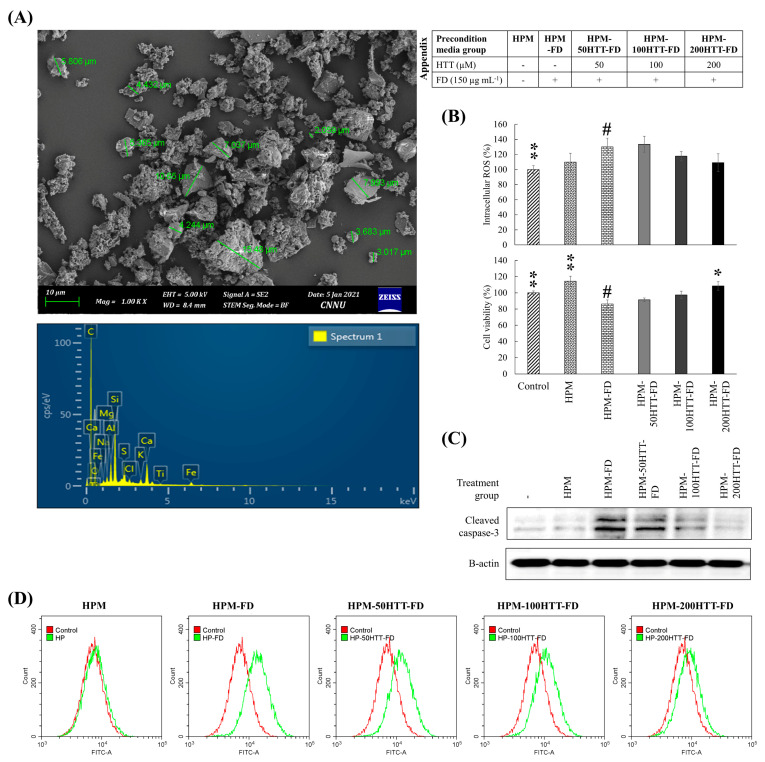
Effect of HTT priming in FD-stimulated HaCaT keratinocyte preconditioned media on producing ROS in integrated HDFs and cytotoxicity. (**A**) SEM secondary electron image of the particle size distribution of FD and SEM-EDX analysis of FD. (**B**) Intracellular ROS level and viability of HDFs. (**C**) Levels of cleaved caspase-3. (**D**) Evaluation of intracellular ROS levels in HDFs by flow cytometry. HDFs were stimulated for 2 h with preconditioned media from FD-stimulated HaCaT keratinocytes with and without HTT pre-treatment. Intracellular ROS levels were measured 2 h after the stimulation period. Cell viability was evaluated after 24 h. Evaluations were carried out in triplicates (*n* = 3) and indicated as means ± SD. “*” and “**” respectively denote *p* < 0.05 and *p* < 0.01 if they are significantly different from those of HPM-FD treated HDFs “#”.

**Figure 2 antioxidants-10-00675-f002:**
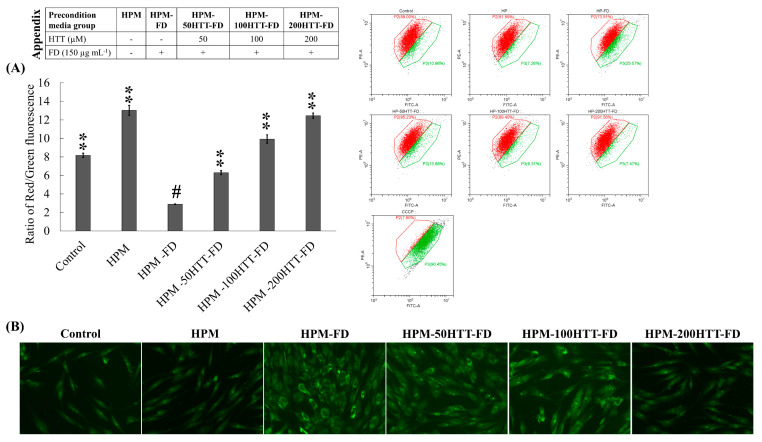
Effect of HTT priming in FD-stimulated HaCaT keratinocyte preconditioned media on mitochondria depolarization and HDF differentiation. (**A**) Mitochondria depolarization as assessed by flow cytometric JC-1 assay. (**B**) Evaluation of nuclear morphology and HDF differentiation by nuclear double staining. Evaluations were carried out in triplicates (*n* = 3) and indicated as means ± SD. “**” respectively denote *p* < 0.01 if they are significantly different from those of HPM-FD treated HDFs “#”.

**Figure 3 antioxidants-10-00675-f003:**
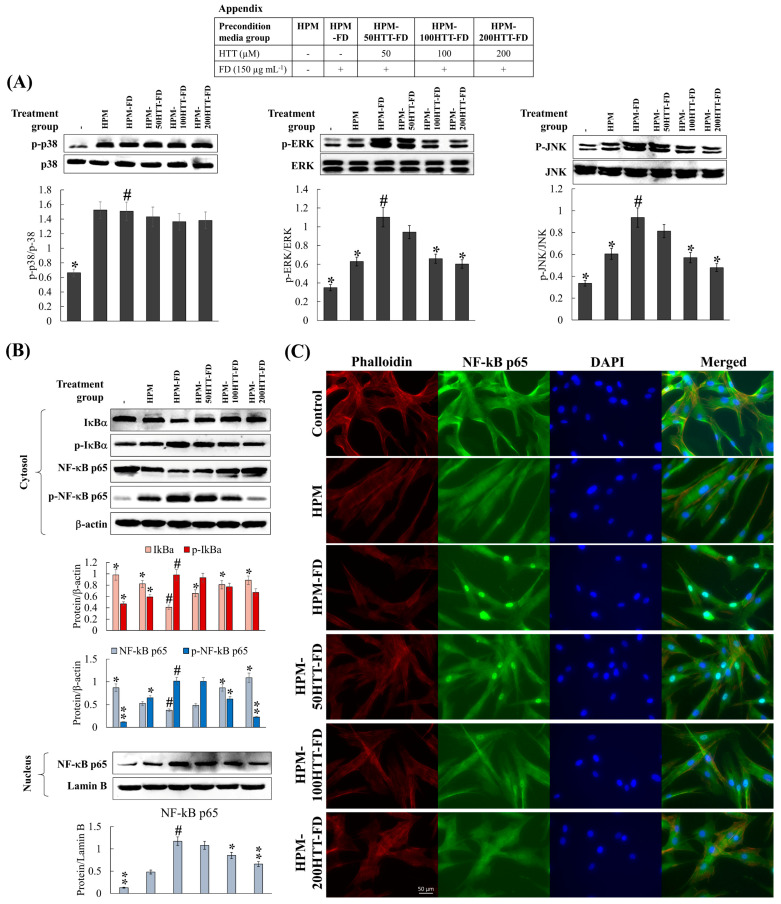
Effect of HTT priming in FD-stimulated HaCaT keratinocyte preconditioned media on MAPK derived NF-κB inflammatory signaling in integrated HDFs. Levels of molecular mediators were assessed by western blot analysis for (**A**) MAPK and (**B**) NF-κB. (**C**) Evaluation of NF-κB p65 nuclear translocation by immunofluorescence analysis. Evaluations were carried out in triplicates (*n* = 3) and indicated as means ± SD. “*” and “**” respectively denote *p* < 0.05 and *p* < 0.01 if they are significantly different from those of HPM-FD treated HDFs “#”.

**Figure 4 antioxidants-10-00675-f004:**
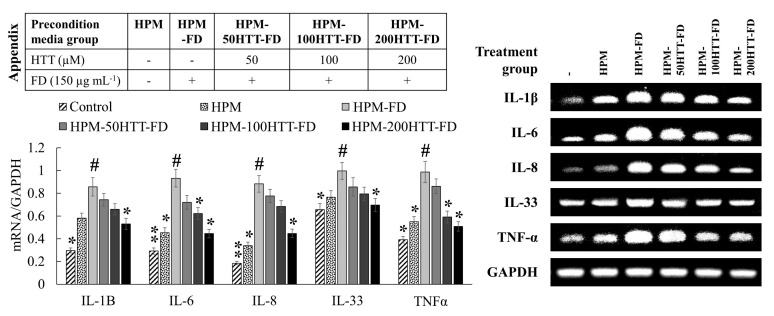
Effect of HTT priming in FD-stimulated HaCaT keratinocyte preconditioned media on inflammatory cytokine expression by integrated HDFs. Evaluations were carried out in triplicates (*n* = 3) and indicated as means ± SD. “*” and “**” respectively denote *p* < 0.05 and *p* < 0.01 if they are significantly different from those of HPM-FD treated HDFs “#”.

**Figure 5 antioxidants-10-00675-f005:**
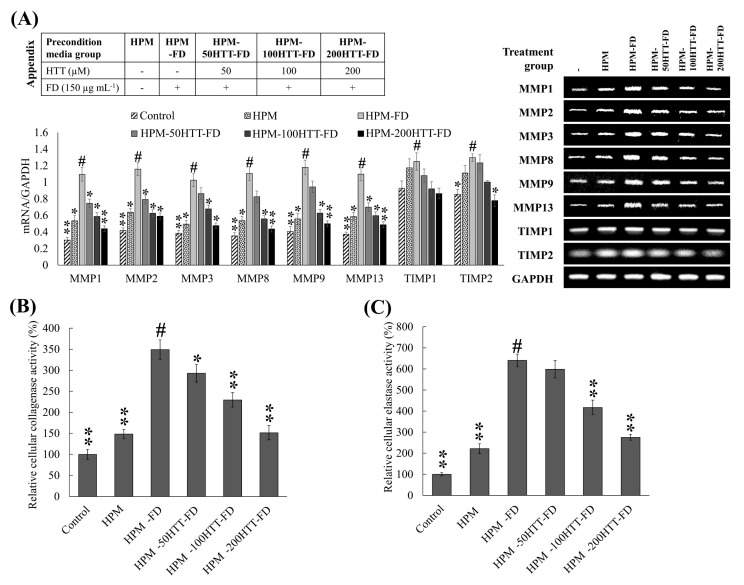
Effect of HTT priming in FD-stimulated HaCaT keratinocyte preconditioned media on ECM degradation by integrated HDFs. (**A**) MMP and TIMP expression was evaluated by RT-PCR analysis. The activity of collagenase (**B**) and elastase (**C**) was evaluated in cell lysates. All experiments were carried out in triplicates (*n* = 3) and indicated as means ± SD. “*” and “**” respectively denote *p* < 0.05 and *p* < 0.01 if they are significantly different from those of HPM-FD treated HDFs “#”.

**Table 1 antioxidants-10-00675-t001:** Abbreviations used for denoting preconditioned media groups.

Treatment	Preconditioned Media Group				
	HPM-Control	HPM-FD	HPM-50HTT-FD	HPM-100HTT-FD	HPM-200HTT-FD
HTT (µM)	-	-	50	100	200
FD (150 µg mL^−1^)	-	+	+	+	+

**Table 2 antioxidants-10-00675-t002:** Forward and reverse primer sequences for PCR analysis.

Target Gene		Primer Sequence (5′ to 3′ Direction)
**IL-1β**	Forward	TGT CCT GCG TGT TGA AAG ATG A
	Reverse	CAG GCA GTT GGG CAT TGG TG
**IL-6**	Forward	GAT GGC TGA AAA AGA TGG ATG C
	Reverse	TGG TTG GGT CAG GGG TGG TT
**IL-8**	Forward	ACA CTG CGC CAA CAC AGA AAT TA
	Reverse	CAG GCA GTT GGG CAT TGG TG
**IL-33**	Forward	GAT GAG ATG TCT CGG CTG CTT G
	Reverse	AGC CGT TAC GGA TAT GGT GGT C
**TNF-α**	Forward	GGC AGT CAG ATC ATC TTC TCG AA
	Reverse	GAA GGC CTA AGG TCC ACT TGT GT
**MMP1**	Forward	CTGAAGGTGATGAAGCAGCC
	Reverse	AGTCCAAGAGAATGGCCGAG
**MMP2**	Forward	GCGACAAGAAGTATGGCTTC
	Reverse	TGCCAAGGTCAATGTCAGGA
**MMP3**	Forward	CTCACAGACCTGACTCGGTT
	Reverse	CACGCCTGAAGGAAGAGATG
**MMP8**	Forward	ATGGACCAACACCTCCGCAA
	Reverse	GTCAATTGCTTGGACGCTGC
**MMP9**	Forward	CGCAGACATCGTCATCCAGT
	Reverse	GGATTGGCCTTGGAAGATGA
**MMP13**	Forward	CTATGGTCCAGGAGATGAAG
	Reverse	AGAGTCTTGCCTGTATCCTC
**TIMP1**	Forward	AGAGTGTCTGCGGATACTTCC
	Reverse	CCAACAGTGTAGGTCTTGGTG
**TIMP2**	Forward	AAGCGGTCAGTGAGAAGGAAG
	Reverse	GGGGCCGTGTAGATAAACTCTAT
**GAPDH**	Forward	CGT CTA GAA AAA CCT GCC AA
	Reverse	TGA AGT CAA AGG AGA CCA CC-

## Data Availability

The data presented in this study are available on request from the corresponding author.

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
