# Peer review of "Effects of (–)-Loliolide against Fine Dust Preconditioned Keratinocyte Media-Induced Dermal Fibroblast Inflammation"

_antioxidants, 2021, doi:10.3390/antiox10050675_

Round 1
Reviewer 1 Report
All comments have been addressed.
Reviewer 2 Report
I have no comments now
This manuscript is a resubmission of an earlier submission. The following is a list of the peer review reports and author responses from that submission.
Round 1
Reviewer 1 Report
This is an interesting study examining the effect of loliolide in fine dust-preconditioned media of dermal fibroblasts. This was interesting and may be of therapeutic interest. The science is sound and reasonably well presented. I have a few comments:
Cell viability is shown to be altered in going down and restored with loliolide treatment- this is a weak way to show cell death and very crude. Active caspase 3 the cleaved caspase- must be shown- this is simply a must
The authors demonstrate at the PCR level increased expression of MMPs, thus an increase in breakdown one would assume, but the authors so not measure the levels of the inihbitors of MMPs- the TIMPs- if their is also a decrease in this the MMPS to TIMP ratio is the same and thus no difference. The authors must measure the expression at lease by PCR of TIMP1 and TIMP2.
The authors must also show a western blot (protein) of collagen 1- which is the main output of MMP regulation.
reference 25 must be supplemented with Ref:
Tumor necrosis factor-costimulated T lymphocytes from patients with systemic sclerosis trigger collagen production in fibroblasts 2013 65: 481-91
Author Response
Comments and suggestions for authors.
This is an interesting study examining the effect of loliolide in fine dust-preconditioned media of dermal fibroblasts. This was interesting and may be of therapeutic interest. The science is sound and reasonably well presented. I have a few comments:
Response: Thank you very much for your comprehensive review and salient comments. We have answered all your comments and modified our manuscript accordingly. Our sincere hope is that this revised article would meet journal requirements for publication. The revised content is marked in red color within the manuscript. Having mentioned that, we would cordially accept any additional comments regarding the revised article.
Comment 1
Cell viability is shown to be altered in going down and restored with loliolide treatment- this is a weak way to show cell death and very crude. Active caspase 3 the cleaved caspase- must be shown- this is simply a must.
Response:
Greatly appreciate your comment. Herein the cell viability was evaluated by the standard MTT assay. We have a set of additional data related to the regulation of intracellular ROS level and apoptosis including the proposed caspase signaling. We were hoping to publish those outcomes in a separate manuscript after completing few more trials. We think that publishing them in here would be too bulky and deviate from the main purpose, which is related to the “skin aging via inflammation”.
However, based on your comments we indicated just the results of cleaved caspase-3. A statement as follows was incorporated explaining the role of caspase-3 in mediating apoptotic cell death.
Please refer to line 239
“Caspases are among the crucial mediators of apoptosis [17]. Caspase-3 is among the most frequently activated protease that catalyzes the cleavage of numerous cellular proteins critical in maintaining cellular homeostasis. Hence analyzing its levels provide vital evidence regarding the loss of cell viability. According to Figure 1C, both control and HPM groups indicated the same level of cleaved caspase-3. A minor increase in cleaved caspase-3 level was observed upon the treatment of HPM-FD. Treatment of preconditioned media from HTT primed HaCaT cells indicated a dose-dependent reduction of cleaved caspase 3.”
Please refer to line 400
“Cleaved caspase-3 is a hallmark of apoptosis. Its extensive involvement in mediating chromatin condensation and DNA fragmentation plays a vital function during normal development while unwanted cells are committed to dying [17]. However, dysregulated caspase-3 cleavage is undesirable as it leads to the loss of cell viability. Results indicated a similar level of cleaved caspase-3 in both the control and HPM treated group. The minor increase of cleaved caspase-3 levels in HDFs treated with HPM-FD relates to the reduction of cell viability in the same group.”
Comment 2
The authors demonstrate at the PCR level increased expression of MMPs, thus an increase in breakdown one would assume, but the authors so not measure the levels of the inhibitors of MMPs- the TIMPs- if their is also a decrease in this the MMPS to TIMP ratio is the same and thus no difference. The authors must measure the expression at lease by PCR of TIMP1 and TIMP2.
Response: Thank you very much for this excellent suggestion. The levels of TIMP1 and TIMP2 play an important part of the MMP signaling. We promptly ordered the primers and carried out PCR. The delay for submission of this revision is due to the above experiments. The results of the TIMP1 and TIMP2 are attached in the manuscript. In addition, we incorporated relevant background information and a discussion of results to the manuscript.
Please refer to the abstract (line 24)
The above events led to cell differentiation and production of matrix metalloproteinases (MMP), increasing collagenase and elastase activity despite the increase of tissue inhibitors of metalloproteinases (TIMP).
Please refer to Table 2 (line 194) Table 2. Forward and reverse primer sequences for PCR analysis.
Please refer to line 331
MMP activities in fibroblasts are regulated by numerous proteins. Among them are the tissue inhibitors of metalloproteinases (TIMPs) [19]. The balance between TIMPs and MMPs is crucial for maintaining the physiological functionalities of ECM. Per transcriptional analysis, a minor increase was seen for TIMP1 and TIMP2 in HPM-FD treated group compared to the control. HDFs treated with HPM-HTT-FD indicated a reduction of TIMP expression along with the HTT dose. Intracellular collagenase and elastase activity indicated a slight upregulation with HPM treatment (Figure 5 B and C).
Please refer to line 468
“In addition to cytokines, MMP production in fibroblasts is regulated by numerous proteins including tissue inhibitors of metalloproteinases (TIMPs) and growth factors [19]. TIMPs play a pivotal role as inhibitors of MMPs. Regulating the balance between TIMPs and MMPs is crucial for the physiological functionalities of ECM. As explicitly described by Arpino et al. (2015), TIMP1 is a potent inhibitor of many MMPs excluding MMP-14, -15, -16, -19, and -24 [30]. The function of TIMP2 is related to the inhibition of MMP-2. However, controversial findings suggest that MMP-2 could be an activator of pro-MMP-2. A minor increase was seen for TIMP1 and TIMP2 transcription in HPM-FD treated group compared to the control. It is reported that the regulation of TIMP expression is independent of MMPs. Though the function of TIMP1 and TIMP2 are related to the inhibition of MMP, the increased activity of collagenase and elastase suggest that the activity of MMPs has surpassed the inhibitory effects of TIMPs. HDFs treated with preconditioned media obtained from HTT primed, FD-stimulated HaCaT cells (HPM-HTT-FD) indicated a reduction of TIMP expression along with the HTT dose. According to previous studies cytokines such as IL-1β, IL-6, and TNF-α positively regulate the selective expression of certain MMPs and TIMPs [19]. Similar outcomes were observed during the present analysis where the fluctuation of cytokines (IL-1β, -6, -8, -33, and TNF-α), MMPs, and TIMPs expression was parallel to each other. Herein translational analysis of TIMP and MMP levels together with type 1 collagen would provide a sound conclusion of the ECM degradative effects of FD preconditioned keratinocyte media on dermal fibroblasts and the effects of HTT.”
Comment 3
The authors must also show a western blot (protein) of collagen 1- which is the main output of MMP regulation.
Response: Appreciate your comment and thank you for your ideas. We ordered the antibodies 揅OL1A1 (E8F4L) XP?Rabbit mAb #72026?from Cell Signaling Technology. However according to our local vendor, it will take some substantial amount of time to be delivered. Hence, we produce this revised manuscript without the western blot results of Type 1 collagen before the deadline for manuscript submission of this special issue (30th April).
The evaluation of collagenase activity in cell lysates was herein considered as one of the endpoint experiment as increased collagenase degradation was resulted from the HPM-FD treatment. With your comments we realize that collagenase activity alone is not sufficient to come to a sound conclusion without considering the rate of collagen production. If somehow the collagen production would surpass the increase in collagenase activity the conclusion of this study would be not the same. However, based on our understanding we think that the above scenario is highly unlikely to happen.
Given the importance of the proposed experiment, we will put a public note in the manuscript (after published) linking to outcomes of pending experiments in the newly implemented “Hypothesis” Open-Source Annotation Tools of MDPI.
Kindly note that we indicated a special note to the readers in the manuscript.
Please refer to line 485
Herein translational analysis of TIMP and MMP levels together with type 1 collagen would provide a sound conclusion of the ECM degradative effects of FD preconditioned keratinocyte media on dermal fibroblasts and the effects of HTT.
Comment 4
reference 25 must be supplemented with Ref:
Tumor necrosis factor-costimulated T lymphocytes from patients with systemic sclerosis trigger collagen production in fibroblasts 2013 65: 481-91
Response: Thank you for your comment. The above study that you have mention was indeed interesting. At first, we thought that it indicates a completely oppose idea to the study by Youn et al. (2011) 34, 890-893 and the outcomes reported in the present manuscript. Hence, we thoroughly investigated into this.
According to that study, conditioned media obtained after the activation of CD3+ lymphocytes with CD3/CD28 beads and TNFR-costimulation, increase the expression of type I collagen in fibroblasts. This does not imply that TNFR-stimulation alone increase the production of collagen. Moreover, systemic sclerosis in this manner may have a different effect on the collagen production by fibroblasts.
Above scenario is different from the one described by Youn et al. (2011) and the present manuscript where TNF-α stimulate MMP-9 (type IV collagenase) expression in keratinocytes via the NF-κB pathway. Hence, we think incorporating information from above study would complicate the idea presented in this study.
To support the observations presented here, we incorporated an additional reference, where it shows that MMP-1 and TIMP-1 mRNA expression were markedly increased with IL-6 and TNF-α treatment and remains unchanged with IL-1β.
Please refer to line 437
Another study indicates that IL-6 and TNF-alpha are implicated in the regulation of TIMP-1 and MMP-1 mRNA expression [19].
Reference
Dasu, M.R.K.; Barrow, R.E.; Spies, M.; Herndon, D.N. Matrix metalloproteinase expression in cytokine stimulated human dermal fibroblasts. Burns 2003, 29, 527-531.
Reviewer 2 Report
The paper by Fernando et al. reports a study about the possible employ of (-)-loliolide (HTT), a natural plant-derived molecule which is renowned for its ethnopharmacological significance and beneficial health properties including antioxidant, anti-inflammatory, anti-allergic, astringent, antipyretic, and vasodilatory activities, in ameliorating intracellular ROS level and inflammatory responses in human dermal fibroblast (HDFs), suppressing oxidative stress and inflammatory responses when these cells are exposed to fine dust.
Fine dust is an etherogenous suspension of solid and liquid particles in the air which chemical composition includes sulfates, nitrates, elemental carbon, organic compounds such PAH, allergens, endotoxins, microbes, cell fragments, and metallic ions/elements (e.g., lead, iron, copper, zinc, nickel, and vanadium). It can penetrate the skin through hair follicles inducing the ROS production and the oxidative stress with subsequent production of inflammatory cytokines, chemokines, and other chemical mediators, resulting in several major skin diseases, including atopic dermatitis, acne, and psoriasis.
The scope of the research developed in this paper was to evaluate whether HTT treatment could modulate the onset and progression of different inflammatory patterns and mechanisms, thus preventing or retarding the above mentioned skin disorders.
Operative strategies and data analysis methods used in the study are clearly presented in the materials and methods section.
The results are discussed in a proper explicative and detailed way. In my opinion results achieved in this study are very satisfying considering the starting point of this work: HTT treatment attenuated cell viability and ROS production in HDFs, attenuated mitochondria depolarization and fibroblast differentiation(responsible for the early stage of apoptosis), suppressed MAPK derived NF-kB inflammatory signalling (two of the main critical upstream pathways responsible for the onset 270 of inflammatory responses), downregulated inflammatory cytokines and chemokines, inhibited connective tissue degradation.
The results summarized above are discussed separately and with the necessary clarifications and statistical reports. The provided supporting material is well prepared.
The manuscript is well structured, clear in the presented topic and enough complete for the references therein quoted. The topic is very new and, in my opinion, it can be of general interest for the broad readership of Antioxidants and, particularly, for those researchers that are keen on studying new strategies of prevention and treatment of skin diseases specifically due to an oxidative/inflammatory response.
On the basis of these consideration I am pleased to recommend this paper for publication in Antioxidants after minor revisions. There are indeed only a few typos and grammar mistakes that should be revised:
- Line 37: after “carbon” there should be a comma instead of a semicolon
- Line 62: “of” seems to be non appropriate there.
- Line 69-72: “Thought this... ... to fibroblast” maybe this sentence could be redrafted in order to be more immediately understandable.
- Line 108: We think that the caption “Table may have a footer” under Table 1 could be an oversight.
Author Response
Comments and suggestions for authors.
The paper by Fernando et al. reports a study about the possible employ of (-)-loliolide (HTT), a natural plant-derived molecule, which is renowned for its ethnopharmacological significance and beneficial health properties including antioxidant, anti-inflammatory, anti-allergic, astringent, antipyretic, and vasodilatory activities, in ameliorating intracellular ROS level and inflammatory responses in human dermal fibroblast (HDFs), suppressing oxidative stress and inflammatory responses when these cells are exposed to fine dust.
Fine dust is a heterogenous suspension of solid and liquid particles in the air which chemical composition includes sulfates, nitrates, elemental carbon, organic compounds such PAH, allergens, endotoxins, microbes, cell fragments, and metallic ions/elements (e.g., lead, iron, copper, zinc, nickel, and vanadium). It can penetrate the skin through hair follicles inducing the ROS production and the oxidative stress with subsequent production of inflammatory cytokines, chemokines, and other chemical mediators, resulting in several major skin diseases, including atopic dermatitis, acne, and psoriasis.
The scope of the research developed in this paper was to evaluate whether HTT treatment could modulate the onset and progression of different inflammatory patterns and mechanisms, thus preventing or retarding the above-mentioned skin disorders.
Response: Thank you very much for the elaborative description and summary with nicely written sentences. We abstracted parts of your sentences and incorporated with the introduction. Also thank you for your comprehensive review and salient comments. We have answered all your comments and modified our manuscript accordingly. Our sincere hope is that this revised article would meet journal requirements for publication. The revised content is marked in red color within the manuscript. Having mentioned that, we would cordially accept any additional comments regarding the revised article.
Comment 1
Operative strategies and data analysis methods used in the study are clearly presented in the materials and methods section.
Response: Thank you for the comment
Comment 2
The results are discussed in a proper explicative and detailed way. In my opinion results achieved in this study are very satisfying considering the starting point of this work: HTT treatment attenuated cell viability and ROS production in HDFs, attenuated mitochondria depolarization and fibroblast differentiation(responsible for the early stage of apoptosis), suppressed MAPK derived NF-kB inflammatory signaling (two of the main critical upstream pathways responsible for the onset of inflammatory responses), downregulated inflammatory cytokines and chemokines, inhibited connective tissue degradation.
Response: Thank you for the comment.
Comment 3
The results summarized above are discussed separately and with the necessary clarifications and statistical reports. The provided supporting material is well prepared.
Response: Thank you for the comment.
Comment 4
The manuscript is well structured, clear in the presented topic and enough complete for the references therein quoted. The topic is very new and, in my opinion, it can be of general interest for the broad readership of Antioxidants and, particularly, for those researchers that are keen on studying new strategies of prevention and treatment of skin diseases specifically due to an oxidative/inflammatory response.
Response: Thank you very much for the comment.
Comment 5
On the basis of these consideration, I am pleased to recommend this paper for publication in Antioxidants after minor revisions. There are indeed only a few typos and grammar mistakes that should be revised:
Response: Thank you for the comment. Once again, we carefully went through the entire manuscript and revised the mistakes. All edited parts are marked in red color text.
Comment 6
Line 37: after “carbon” there should be a comma instead of a semicolon.
Response: Thank you. We revised the typo.
Please refer to line 37
“The chemical composition of FD includes sulfates, nitrates, elemental carbon, organic compounds,,,”
Comment 7
Line 62: “of” seems to be non-appropriate there.
Response: Yes, you are correct. Thank you and we revised the issue.
Please refer to line 62
“Although these studies formed the foundation of current knowledge, they are not suitable to imitate the interactions between different cell types, especially the spatial association between skin cell layers.”
Comment 8
Line 69-72: “Thought this... ... to fibroblast” maybe this sentence could be redrafted in order to be more immediately understandable.
Response: Appreciate your comment. Yes, it will better if the sentence is clearer. Hence, we modified the statement.
Please refer to line 62
This method provides the ability to monitor the transfer of signals from keratinocytes to fibroblasts. However, this integrated co-culture technique is not a putative method as it lacks several important cues of fibroblast and keratinocyte environment, such as extracellular matrix (ECM), stiffness, much lower cell density compared to the use of in vitro and organotypic skin culture models.
Comment 9
We think that the caption “Table may have a footer” under Table 1 could be an oversight.
Response: Thank you. Seems we have forgotten to remove it from the template. Note that we corrected the issue.